# Polydatin Incorporated in Polycaprolactone Nanofibers Improves Osteogenic Differentiation

**DOI:** 10.3390/ph15060727

**Published:** 2022-06-08

**Authors:** Stefania Lama, Amalia Luce, Giuseppe Bitti, Pilar Chacon-Millan, Annalisa Itro, Pasquale Ferranti, Giovanni D’Auria, Marcella Cammarota, Giovanni Francesco Nicoletti, Giuseppe Andrea Ferraro, Chiara Schiraldi, Michele Caraglia, Evzen Amler, Paola Stiuso

**Affiliations:** 1Department of Precision Medicine, University of Campania “Luigi Vanvitelli”, 80138 Naples, Italy; stefania.lama@unicampania.it (S.L.); amalia.luce@unicampania.it (A.L.); pilar.chaconmillan@unicampania.it (P.C.-M.); michele.caraglia@unicampania.it (M.C.); 2Institute of Biophysics, 2nd Faculty of Medicine, Charles University, V Uvalu 84, 15006 Prague, Czech Republic; giuseppebitti44@gmail.com (G.B.); evzen.amler@lfmotol.cuni.cz (E.A.); 3Plastic Surgery Unit, Department of Multidisciplinary Medical and Dental Specialties, University of Campania “Luigi Vanvitelli”, 80138 Naples, Italy; annalisa.itro@unicampani.it (A.I.); giovannifrancesco.nicoletti@unicampania.it (G.F.N.); giuseppe.ferraro@unicampania.it (G.A.F.); 4Department of Agricultural Sciences, University of Naples Federico II, 80138 Portici, Italy; ferranti@unina.it (P.F.); giovanni.dauria@unina.it (G.D.); 5Department of Experimental Medicine, Section of Biotechnology, Molecular Medicine and Medical Histology, University of Campania “L. Vanvitelli”, 80138 Naples, Italy; marcella.cammarota@unicampania.it (M.C.); chiara.schiraldi@unicampania.it (C.S.)

**Keywords:** osteosarcoma, polydatin, osteogenic differentiation, mesenchymal stem cells, polycaprolactone nanofibers

## Abstract

Polycaprolactone nanofibers are used as scaffolds in the field of tissue engineering for tissue regeneration or drug delivery. Polycaprolactone (PCL) is a biodegradable hydrophobic polyester used to obtain implantable nanostructures, which are clinically applicable due to their biological safety. Polydatin (PD), a glycosidic precursor of resveratrol, is known for its antioxidant, antitumor, antiosteoporotic, and bone regeneration activities. We aimed to use the osteogenic capacity of polydatin to create a biomimetic innovative and patented scaffold consisting of PCL-PD for bone tissue engineering. Both osteosarcoma cells (Saos-2) and mesenchymal stem cells (MSCs) were used to test the in vitro cytocompatibility of the PD-PCL scaffold. Reverse-phase (RP) HPLC was used to evaluate the timing release of PD from the PCL-PD nanofibers and the MTT assay, scanning electron microscopy, and alkaline phosphatase (ALP) activity were used to evaluate the proliferation, adhesion, and cellular differentiation in both osteosarcoma and human mesenchymal stem cells (MSCs) seeded on PD-PCL nanofibers. The proliferation of osteosarcoma cells (Saos-2) on the PD-PCL scaffold decreased when compared to cells grown on PLC nanofibers, whereas the proliferation of MSCs was comparable in both PCL and PD-PCL nanofibers. Noteworthy, after 14 days, the ALP activity was higher in both Saos-2 cells and MSCs cultivated on PD-PCL than on empty scaffolds. Moreover, the same cells showed a spindle-shaped morphology after 14 days when grown on PD-PCL as shown by SEM. In conclusion, we provide evidence that nanofibers appropriately coated with PD support the adhesion and promote the osteogenic differentiation of both human osteosarcoma cells and MSCs.

## 1. Introduction

Osteosarcoma (OS) is the most common malignant tumor among skeletal cancers [1]. OS presents a bimodal age distribution, showing the first peak in adolescents and young adults between the ages of 10 and 25 years and the second peak in older adults (after 65 years of age) [2]. Pediatric OS occurs in children and young adults with a greater incidence, suggesting a correlation with the adolescent growth spurt [3,4,5]. On the contrary adult osteosarcoma is more likely to represent a secondary malignancy and is frequently related to Paget’s disease. The propensity of OS in adolescents indicates a close correlation with rapid bone proliferation. Recently, mesenchymal stem cells (MSCs) have emerged as having a key role in OS transformation and progression [6]. MSCs are multipotent stromal cells with self-renewal abilities and a capability to differentiate into osteoblasts, chondrocytes, and adipocytes. In this light, OS has been reconsidered as a differentiation disease that is caused by genetic and epigenetic alterations, which may impair normal bone development by blocking multipotent MSCs’ differentiation into osteoblasts. Cancer stem cells (CSCs) with typical features of pre-osteoblastic MSCs have been discovered in OS; their presence in OS has been hypothesized to explain their tumor heterogeneity, chemotherapy resistance, and high capacity to metastasize [7]. Therapies for OS include neoadjuvant chemotherapy, surgery, and adjuvant chemotherapy; however, the 5-year survival rate for patients diagnosed with osteosarcoma remains at 60–75% [8]. Nanomaterials are used in the field of tissue engineering as scaffolds for tissue regeneration or the delivery of active agents [9,10]. Polycaprolactone (PCL) is a biodegradable hydrophobic polyester approved by the Food and Drug Administration for medical use to obtain implantable nanostructures that are biologically safe and clinically applicable [11]. Sonomoto et al. demonstrated that the transplantation of MSCs onto a poly-lactic-co-glycolic acid nanofiber scaffold could potentially inhibit the progression of bone destruction and simultaneously promote bone regeneration in patients with uncontrollable arthritis [12]. Moreover, the incorporation of molecules with bone-regenerating activities in PCL nanofibers creates biomimetic scaffolds for bone tissue engineering. Polydatin (PD) is a molecule that was originally isolated from the *Polygonum cuspidatum* plant, and is known for its antioxidant, anti-inflammatory [13], and antitumor effects [14,15]. Our previous studies suggested that PD shifts undifferentiated Caco-2 cells to differentiated enterocytes, with the redistribution of vimentin ending in a process of programmed cell death [16]. Moreover, PD sensitized OS cells to ionizing radiation and this effect was paralleled by increased expression and secretion of ceramides and sphingolipids [17]. PD was recently found to alleviate osteoporosis symptoms in the ovariectomized (OVX) mouse model by upregulating the expression of β-catenin [18]. Previously, it was demonstrated that PD improved the osteogenic differentiation of MSCs and maintained the bone matrix in the OVX mouse model through the activation of TAZ, a potential target gene of the BMP2-Wnt/β-catenin pathway [19]. In this context, we used electrospun nanofibrous tissue scaffolds coated with polydatin to improve the adhesion and osteogenic differentiation of both human osteosarcoma and MSCs. The present manuscript focuses on an innovative way to deliver and release a natural product such as PD in the bone microenvironment for both anticancer and regenerative purposes.

## 2. Results 

### 2.1. Characterization of the PD-PCL Scaffold 

The PD-PCL nanofibers were morphologically characterized by scanning electron microscopy (SEM) while the concentration and time release of PD from the scaffolds were evaluated by RT-HPLC. Figure 1 shows the surfaces of the PCL and PD-PCL nanofibers obtained from the SEM analysis. PD-PCL nanofibers (Figure 1A) showed the presence of numerous agglomerates (yellow arrows) compared to the control sample (empty PCL). The PD release from PD-PCL was analyzed in different cell growth media. In Figure 2A, we report the representative spectra of the samples resulting from the incubation of PCL and PD-PCL nanofibers in DMEM compared to a PD standard solution. The RP-HPLC of the PD-PCL nanofibers in DMEM (Figure 2) highlights the presence of a peak after around 30 min, which is confirmed to be PD by the concomitant chromatography of the standard PD. Therefore, in Figure 2 (panel D), we show the PD concentration released from the PD-PCL nanofibers under different experimental conditions. Noteworthy, after 7 days of incubation, the amount of PD in the DMEM medium reached a plateau of about 39 µM (Figure 2). The best performance regarding the PD release is reached when DMEM and α-MEM media were applied to the nanofibers. This result may be explained by the different solubility of PD in the different buffers. The SEM and RT-HPLC analysis suggested the formation of hydrogen bonds between the free hydroxyl groups of the PD and the carbonylic groups of the polymeric electrospun scaffolds (Figure 1B). The FI-TR preliminary spectra of PCL and PD-PCL did not detect putative hydrogen bonds (data not shown) due to the low PD concentration. The interconnectivity in PD-PCL, as evidenced by SEM, was an important factor for cell stratification into scaffolds. These observations suggest that these scaffolds are morphologically suitable for cell adhesion.

### 2.2. PD-PCL Scaffold Induced Differentiative Effects on Saos-2 and MSCs

We examined the cell proliferation and alkaline phosphatase activity (ALP) of both human OS (Saos-2) and human bone mesenchymal stem cells (MSCs) for a period of 1, 7, and 14 days on PD-PCL nanofibers to investigate the biocompatibility of scaffolds incorporating PD (Figure 3). After 14 days, the proliferation of Saos-2 cells on PD-PCL was significantly decreased by about 25% (*p* = 0.023) compared to Saos-2 cells grown on PCL scaffolds. On the other hand, no effect on the proliferation of MSCs grown on PD-PCL compared to cells grown on PCL alone was detected. These results indicate that the slow release of PD from the PD-PCL scaffolds induced cell toxicity in Saos-2 cells while supporting MSC cell proliferation. Subsequently, we evaluated the activity of ALP, up to 14 days, both in Saos-2 and MSC cells (Figure 3). ALP activity was significantly increased in both Saos-2 cells and MSCs cultured on PD-PCL compared to cells grown on the empty scaffold. OS is an aggressive bone cancer that occurs mainly in adolescents [1], with a highly metastatic form, principally in the lungs, induced by the mesenchymal component. In our previous research, we highlighted an antiproliferative effect of PD on Saos2 cells, with an IC_25_ of 48 μM, together with a reduction in invasion and cell migration [17]. Here, after 14 days, we recorded that PD was released from the PCL scaffolds at a concentration of about 40 μM, which was paralleled by an antiproliferative effect and a concomitant increase in ALP activity in Saos2 cells grown on PD-PCL. These results are in line with our previously reported findings [17]. Noteworthy, PD-PCL scaffolds were not toxic to MSC cells, where they induced differentiation after 14 days of incubation.

### 2.3. Polydatin Promoted Morphological Osteogenic Differentiation of Saos-2 and MSCs

The morphological change and differentiation induced in Saos-2 cells seeded on the PD-PCL scaffold were evaluated after 14 days by scanning electron microscopy (SEM). As shown in Figure 4, differences in cell morphology were detected between Saos-2 cells adhered to a scaffold composed of PCL, PD-PCL, and PCL + PD (48 μM). Saos-2 cells were relatively few and sparse when grown on PD-PCL nanofibers compared to the Saos-2 cells grown on empty scaffolds. When the Saos-2 cells were cultured on PCL, the PD added to the medium showed evident polygonal long processes. This finding could be due to the amount of PD required to induce differentiation in a model of human OS. The MSCs on PCL scaffolds were rounded, suggesting poor cell adhesion; however, when cultured on PD-PCL, the cells appeared elongated with cytoplasmic extroversions, suggesting osteoblastic differentiation (Figure 5). These findings were supported by the increase in the production and deposition of collagen fibers in the scaffolds when MSCs were grown on PD-PCL compared to the cells incubated on both PCL and PCL + PD (Figure 6). Contrarily the Saos-2 cells showed more evident differentiation when PD was added directly to the medium.

## 3. Materials and Methods

### 3.1. Chemicals

Cell culture plastics were purchased from Becton Dickinson (Lincoln Park, NJ, USA). Fetal bovine serum (FBS), phosphate-buffered saline (PBS), L-glutamine, trypsin, and antibiotics were purchased from Gibco (Life Technologies, Carlsbad, CA, USA). Trans-polydatin with a purity grade higher than 99% was provided by Glures S.r.l., Academic Spin-Off of Ca’ Foscari University of Venice (Venice, Italy). Minimum Essential Medium Eagle Alpha Modification (α-MEM) was purchased from Gibco, Thermo Fisher Scientific, Inc. (Waltham, MA, USA). All other regents used in this study were of analytical grade. Polycaprolactone was purchased from Sigma-Aldrich (Sigma-Aldrich; Merck KGaA, Darmstadt, Germany). Chloroform and ethanol were purchased from VWR International (VWR International, Radnor, PA, USA).

### 3.2. Nanofiber Electrospinning and Coating with Polydatin 

Nanofibers were prepared at room temperature using the electrospinning method using a ™ NS 1S500U, Elmarco device (Elmarco, Czech Republic), with a maximal voltage of up to 100 kV. The solution for the electrospinning process contained 10% (*w/v*) PCL ((MW 40,000 Wako Chemicals GmbH, Neuss, Germany) dissolved in a mixture of chloroform and ethanol at a 10:1 ratio (VWR International, Radnor, PA, USA). The PCL solution used to perform the polydatin-PCL (PD-PCL) nanofiber contained a 0.5% (*w/v*) polydatin concentration. Electrospinning was performed using a needleless wire electrode. The fibers produced were deposited on a nonwoven supporting textile (Spunbond, Pegas Textiles, Prague, Czech Republic). After production, the nanofiber towels were stored at room temperature, samples of each towel were collected, placed on aluminum stubs, and coated with a gold layer employing a Polaron sputter-coater (SC510, Polaron, Now Quorum Technologies Ltd., Lewes, UK). The samples treated with gold were studied with an electron microscope Aqua SEM (Tescan, Brno-Kohoutovice, Czech Republic). 

### 3.3. Polydatin Release from PD-PCL Nanofibers

The release of PD from coated PCL nanofibers was evaluated by reverse-phase (RP) HPLC. The analysis was carried out using an HP 1100 modular system equipped with a diode array detector (Agilent, Palo Alto, CA, USA). A polydatin standard solution (14 ng/µL) in ethanol was used for quantitative analysis. Extracted samples were diluted 1:2 in ethanol and 50 µL was injected. The separation was performed using a Macherey-Nagel C18 EC 250/4 NUCLEODUR 100–5, 4.6 × 250 mm column (Düren, Germany). Solvent A was 0.1% TFA *v/v* in water; solvent B was 0.1% TFA in acetonitrile *v/v*. After 5 min of isocratic elution using 5% solvent B, a 5–65% gradient ramp was applied over 60 min at a flow rate of 1.0 mL/min. The column effluents were monitored by detection at λ = 306 nm.

### 3.4. Cell Culture

The human osteosarcoma cell line Saos-2 was grown in Iscove’s Modified Dulbecco’s Medium (IMDM) (Sigma-Aldrich; Merck KGaA, Darmstadt, Germany). The medium was enriched with 10% heat-inactivated fetal bovine serum (Sigma-Aldrich; Merck KGaA, Darmstadt, Germany), 1% L-glutamine, and antibiotics (1.0 × 10^4^ U/mL penicillin, 1.0 × 10^4^ µg/mL streptomycin). The cell line was grown at 37 °C in a 5% CO_2_-humidified atmosphere. The Saos-2 cells were plated on PCL and PD-PCL fibers at a density of 8 × 10^4^ cells/cm^2^ and were grown for 14 days in DMEM.

Human mesenchymal stem cells (MSCs) were grown in α-MEM medium supplemented by 15% heat-inactivated fetal bovine serum FBS, 50 mg/L ascorbic acid, and 10 mM β-glycerophosphate. The MSCs were plated at a density of 8 × 10^4^ cells/cm^2^ on PCL and PD-PCL nanofibers and cultured in the osteogenic differentiation medium for 14 days. 

### 3.5. Morphology Analysis (SEM) 

The morphology of the PD-PCL nanofibers was evaluated by SEM. Air-dried PCL and PD-PCL were mounted on aluminum stubs and sputter-coated and a platinum-palladium Denton Vacuum (DESK V). FESEM (Field-Emission SEM) Supra 40 (ZEISS; EHT = 5.00 kV, WD = 22 mm, detector in lens) was used for observation. The mean fiber diameter was calculated by image analysis in the ImageJ program. To visualize the cellular morphology, the nanofibers with and without cells were rinsed with PBS and then fixed with 2.5% glutaraldehyde for at least 1 h and following the ethanol series (35%, 48%, 70%, 96%, 100%, each for 10 min); after the addition of a few drops of HMDS and slow evaporation, the nanofibers were mounted on aluminum stubs and sputter-coated with a layer of gold using a Quorum Q150R. The samples were examined in a Vega 3 SBU (Tescan) SEM in the secondary electron mode at 15 kV.

### 3.6. Alkaline Phosphatase (ALP) Staining

The ALP activity was used to assess the differentiation in Saos-2 and MSC growth on the PCL and PD-PCL scaffold. After 1, 7, and 14 days, the cell layers were extracted by the addition of 500 µL 0.1% Triton-X in 10 mM Tris-HCl (pH 7.4) and incubation at 4 °C on a gyratory shaker for 2 h (see Sigma Technical Bulletin Procedure No.104). ALP activity was evaluated by measuring the p-nitrophenol production during incubation for 15 min at 37 °C, with p-nitrophenyl phosphate as the substrate (1-Step™ PNPP kit, Thermo Fisher Scientific). The absorbance was measured at 405 nmon a Perkin Elmer BioAssay Reader HTS 7000.

### 3.7. Immunofluorescence Staining

On days 7 and 14, the samples were fixed with cold 70% methanol and stored at −20 °C until analysis. After thawing for a few minutes, the samples were washed with PBS and incubated with 1% bovine serum albumin (BSA) in PBS/0.1% (*v/v*) Triton-X for 30 min at RT. After washing with PBS, the primary monoclonal antibody against type I collagen (dilution 1:200, LB-1197, CosmoBio Co., LTD., Tokyo, Japan) was applied overnight at 2–8 °C. The day after, the samples were incubated with Anti-Rabbit IgG Alexa Fluor^®^ 488 secondary antibody for 45 min at RT in the dark (Invitrogen, Life Technologies, Eugene, OR, USA). After washing with PBS, cell nuclei were stained for 5 min with propidium iodide (PI) (RT, 5 mg/mL in PBS, Sigma-Aldrich; Merck KGaA, Darmstadt, Germany). The cells were visualized using a confocal microscope Zeiss LSM 880 Airyscan, (Zeiss, Oberkochen, Germany) at ʎ_exc_ = 488 nm and ʎ_em_ = 505–550 nm for Alexa Fluor 488 and at ʎ_exc_ = 535 and ʎ_em_ > 570 nm for PI.

## 4. Discussion

Surgical resection, chemotherapy, and radiotherapy form the therapeutic strategy of bone tumors; however, about 20% of patients with primary osteosarcoma experience local recurrence or metastasis. Tumor local relapse can be attributed to insufficient resection of tumor, hidden multifocal tumor, and non-responsive tumor post-surgical radiation therapy. Many authors have reported that MSCs can be recruited into the microenvironment of OS [20,21,22] and play an important role in the growth, progression, metastasis, and drug resistance of OS. Furthermore, it was shown in vitro that OS cells induced the epithelial mesenchymal transition of bone mesenchymal stem cells (BMSCs) to cancer-associated fibroblasts and the Notch and Akt signaling pathways mediated the EMT process [23,24,25,26,27,28]. The use of microporous biomaterials complexed with anticancer agents represents a potentially active and easy strategy in both anticancer and regenerative medicine. In fact, biomaterials have an important dual purpose: to release the antitumor drugs in the bone at the place of surgical intervention and to mimic the bone microenvironment by inducing tissue regeneration [20]. Polydatin, a natural glycosylate precursor of resveratrol, has been reported to have an antitumor effect by inhibiting the PI3K/Akt and PDGF/Akt pathway [29,30,31,32]. It was reported that polydatin inhibited the proliferation and promoted the apoptosis of OS cells through upregulation of the Bax/Bcl-2 ratio and attenuation of β-catenin signaling [33]. These effects occur together with a decrease in the cell proliferation, invasion, and inhibition of the EMT process. In our precedent paper, we demonstrated that PD was able to induce cell cycle arrest in the S phase, redistributed β-catenin protein, and enhanced osteoblast differentiation markers, including phosphatase alkalin activity, osteopontin, and Notch2 [17]. In addition, polydatin overcomes resistance to paclitaxel in human OS cell lines [34,35]. Published reports suggest that the use of poly (epsilon-caprolactone) complexed with a variety of natural molecules or polymers enhances its biocompatibility and/or its therapeutic action for bone repair. In the present study, we demonstrated that polycaprolactone nanofibers complexed to polydatin (PD-PCL) were able to stimulate the osteogenic differentiation of both OS and bone human mesenchymal stem cells. The sustained and controlled release of osteogenic and anticancer biomolecules through biodegradable PCL nanofibers for OS bone implants is important for correct osteogenesis. Understanding the PD release time from the PCL polymer is pivotal for the design of a scaffold-based drug delivery device. Using RT-HPLC, we studied the release of PD from PCL in different media such as DMEM, α-MEM, PBS, and RPMI. After 7 days, we observed a higher release of PD (40 μM) in DMEM and α-MEM media compared to the PD release in PBS (27 μM) and RPMI (28 μM) medium. The glucose group in the PD structure represents protection from oxidation and increases the solubility of PD [36], increasing its hydrophilic properties. On the other hand, a microstructure can only form if the molecule is soluble in the PCL matrix. The release of PD occurred in 7 days, and this suggests that the latter is not due to either a diffusion process, which would occur in the first hours, or biodegradation of the PCL fibers. These results confirm that the PD release from PCL fibers is slow and controlled and is due to hydrophilic–hydrophobic interactions that are established between the molecule and polymer. Based on the good uptake and slow release of PD from PD-PCL nanofibers, we evaluated their effects on the cell attachment, proliferation, and osteogenic differentiation of both Saos-2 and MSC cells. Based on these experiments, we found that PD-PCL fibers, although PD is complexed to PCL nanofibers, maintained both antiproliferative and osteogenic differentiating effects on Saos-2 cells as previously demonstrated by our group [17]. It is well known that PCL nanofibers have good biocompatibility and our in vitro MSC cell viability assay confirmed that no differences were observed between the cell growth on PCL compared to the MSC growth on the PD-PCL scaffolds. However, after 14 days, the MSC growth on PD-PCL scaffolds without osteogenic differentiation medium exhibited significantly increased ALP activity compared to the cells on the PCL scaffolds. This might be attributed to both the different physical properties of the PD-PCL scaffolds and to the biological action of PD released in the medium. All these properties indicate that the PD-PCL scaffold could function as an in vivo bone environment, allowing host cells to populate the tissues and specialize into osteoblasts with regenerative capabilities. A limitation of the present study is based on the following observation: the antiproliferative effect of PD was higher when it was added directly to the medium of seeded OS cells. This finding suggests that for in vivo translation, contemporary systemic administration of PD together with nanofiber implantation should be performed. Type I collagen is a component of the bone extracellular matrix that provides physical and mechanical support to tissues. Recently, collagen I scaffolds were used in tissue engineering, including nerve, bone, cartilage, tendon/ligament, vascular grafts, and skin [37,38], which all provided obvious promotion functions regarding tissue repair, both in vitro and in vivo. Interestingly, we evidenced, by immunofluorescence, an increased expression of collagen I after 14 days in MSCs grown on both PD-PCL and PCL + PD compared to MSCs grown on PCL. Increased collagen I expression is very important for providing spatial and mechanical cues to cells and physical support for tissues. In fact, collagen may be an attractant for fibroblasts in vivo during wound repair and fracture healing.

In conclusion, our results strongly suggest that PD-PCL nanofibers can be efficiently used to both sterilize the bone tissue of remaining cancer tumor cells and to provide a useful support for the proliferation and osteogenic differentiation of MSCs.

## Figures and Tables

**Figure 1 pharmaceuticals-15-00727-f001:**
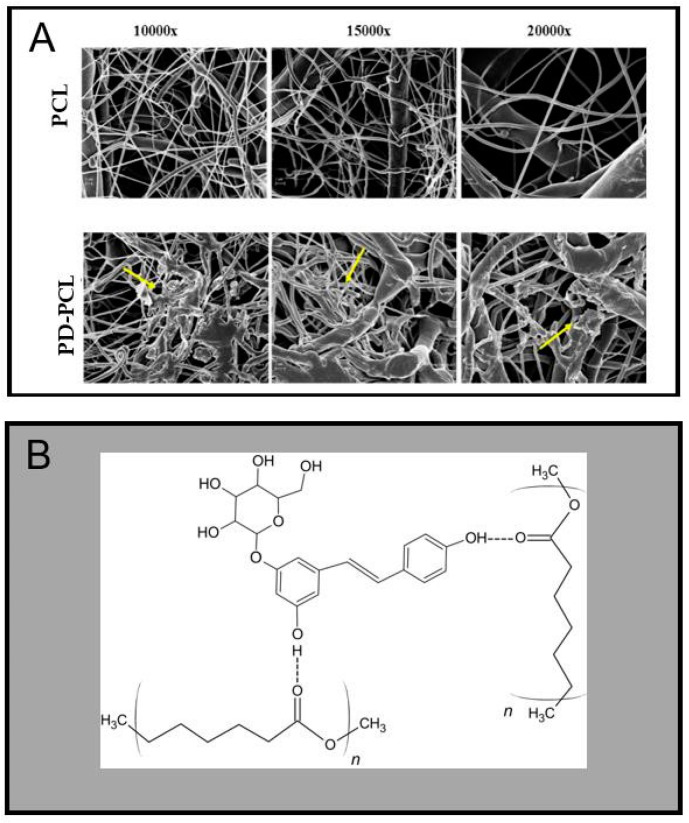
(**A**) Images of polycaprolactone (PCL) and polydatin-incorporated polycaprolactone (PD-PCL) nanofibers analyzed by scanning electron microscopy (SEM). (**B**) Schematic representation of the hypothetical hydrogen bonding interactions between PCL–PD copolymers.

**Figure 2 pharmaceuticals-15-00727-f002:**
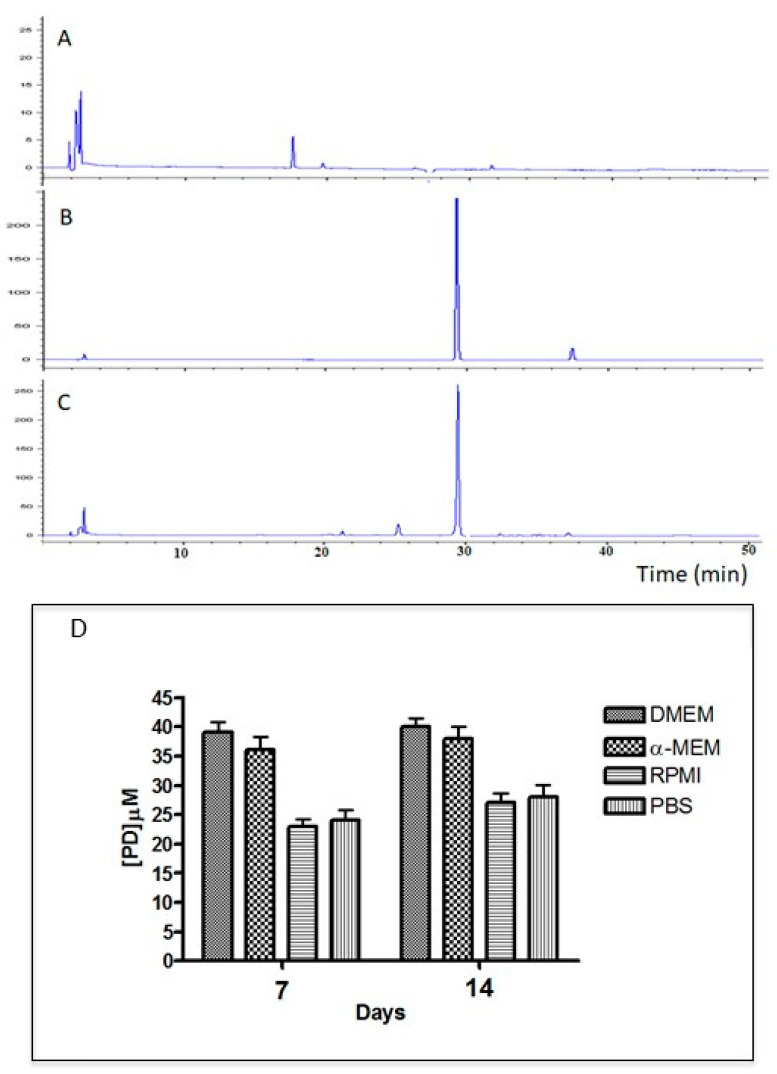
Analysis of the release of PD from PD-PCL nanofibers. (**A**) RT-HPLC spectrum of media (DMEM), (**B**) RT-HPLC spectrum of PD standard (14 ng/µL), and (**C**) RT-HPLC spectrum of PD released in DMEM from the PD-PCL scaffold. The release of PD was sustained for up to 14 days in PD-PCL without a burst effect. (**D**) PD released from PD-PCL after 7 and 14 days of incubation under different experimental conditions as evaluated by RT-HPLC.

**Figure 3 pharmaceuticals-15-00727-f003:**
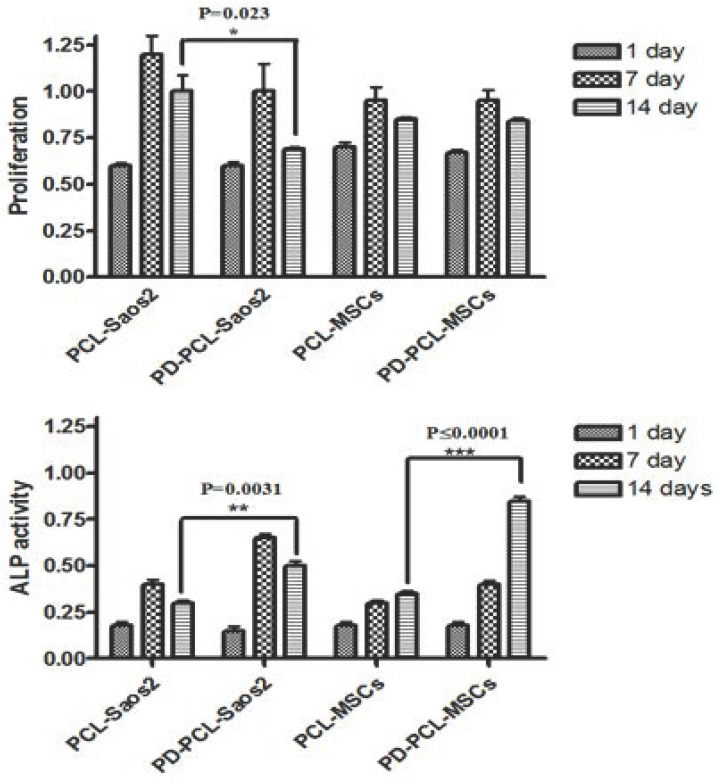
Cell proliferation and alkaline phosphatase activity of Saos-2 and MSC cell lines seeded on both PCL and PD-PCL scaffolds. * *p* ≤ 0.05; ** *p* ≤ 0.005; *** *p* ≤ 0.0001.

**Figure 4 pharmaceuticals-15-00727-f004:**
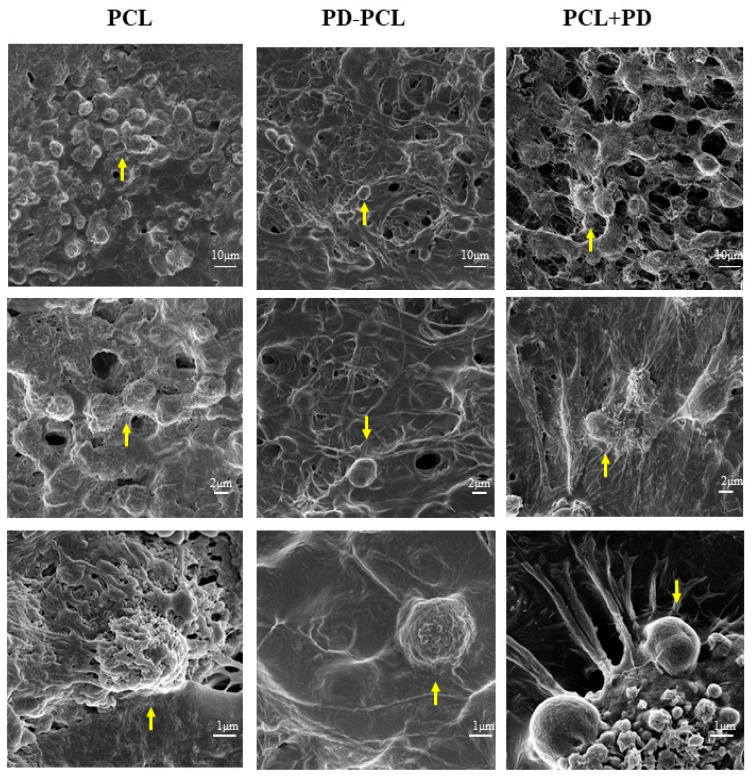
Representative images at different magnifications of Saos-2 cells grown for 14 days on: polycaprolactone scaffold (PCL); PD-coated polycaprolactone (PD-PCL) scaffold; PCL scaffold with 48 μM PD added to the medium (PCL + PD).

**Figure 5 pharmaceuticals-15-00727-f005:**
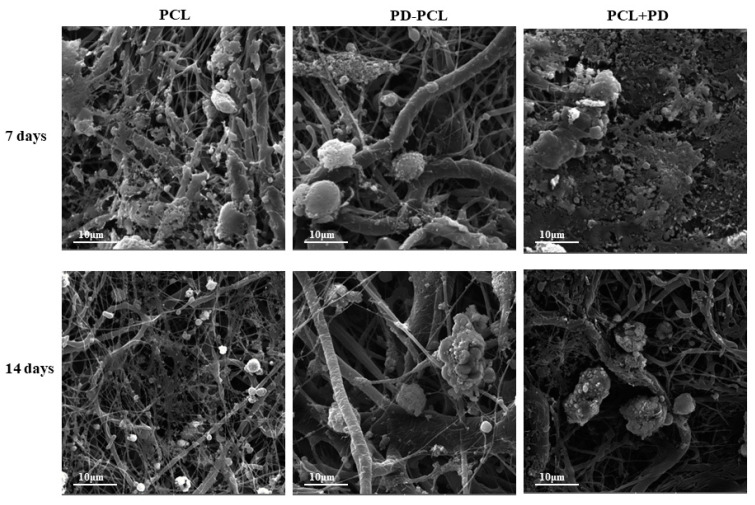
SEM Images of MSC cells grown for 7 and 14 days on: polycaprolactone scaffolds (PCL); PD-coated polycaprolactone scaffolds (PD-PCL); PCL scaffold with 48 μM PD added to the medium (PCL + PD).

**Figure 6 pharmaceuticals-15-00727-f006:**
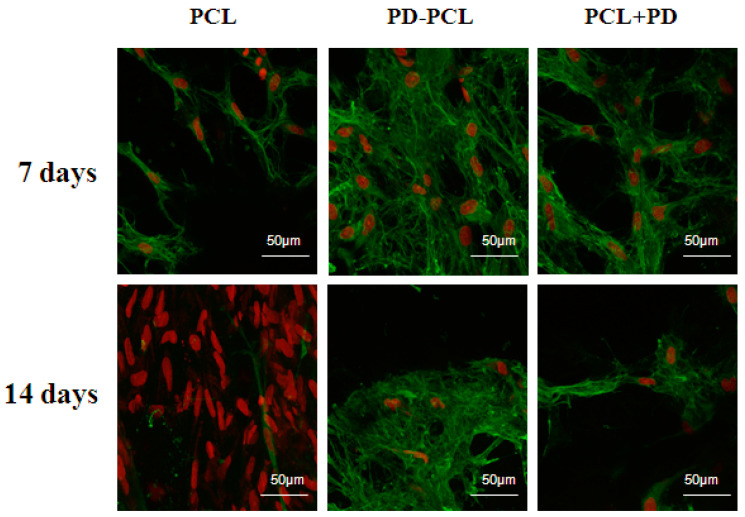
Immunofluorescence analysis of the MSCs cultured on PCL and PD-PCL scaffolds after 7 and 14 days. Representative images of confocal microscopy analyses showing the immunopositivity and distribution of collagen (green) in MSCs cultured on polycaprolactone (PCL); polydatin-coated polycaprolactone (PD-PCL); PCL scaffold with 48 μM PD added to the medium (PCL + PD). The nuclei were PI stained (red).

## Data Availability

Data is contained within article.

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
