# Peer review of "Polydatin Incorporated in Polycaprolactone Nanofibers Improves Osteogenic Differentiation"

_pharmaceuticals, 2022, doi:10.3390/ph15060727_

Round 1

Reviewer 1 Report

The manuscript “Polydatin incorporated in Polycaprolactone nanofibers improves osteogenic differentiation” by Lama et al. reports the osteogenic capacity of polydatin to create a bio-mimetic scaffold consisting of PCL-PD for bone tissue engineering. However, some important results are lack from this manuscript; therefore, I would suggest authors may take a least a major revision of this work. Here are the comments and suggestions:

  1. In Table 1, the accumulation release of PD from PD-PCL should be plotted.
  2. In Fig. 3, the cytotoxicity pathway of the PD-PCL-Saos2 should be studied.
  3. The differentiation pathway of the Saos and MSCs should be studied and discussed.
  4. Scale bars in Figs. 4-6 are unclear.
  5. 5 is dislocated.
  6. In Fig. 6, cell number on PCL seems less than that on PD-PCL on day 7, which is disagreed with results in Fig. 3. Can authors stain both ALP and COL I in the merge image to see the stage of differentiation?
  7. Page 9, how could authors comment that the good uptake and slow release of PD?

Author Response

Response to Reviewer 1

Issue 1. In Table 1, the accumulation release of PD from PD-PCL should be plotted.

Response: In the revised version of the manuscript, we have added into figure 2 panel D where the plotted PD release from PD-PCL in different cellular media is reported, as correctly suggested by the referee.

Issue 2. In Fig. 3, the cytotoxicity and differentiation pathway of the PD-PCL-Saos2 should be studied.

Response: We have studied PD cytotoxic effect mechanisms on Saos-2 cells in a previously published manuscript (Amalia Luce et al. Oxid Med Cell Longev. 2021). In this paper, we demonstrated that PD was able to induce cell cycle arrest in S phase, redistributed β-catenin protein, and reduced clonogenic survival of tumor cells as stated in the “Discussion” section of the revised version of the manuscript.

Issue 3. The differentiation pathway of the Saos and MSCs should be studied and discussed.

Response: We thank the reviewer for her\his kind comment.  We have studied the differentiation pathway of the Saos-2 in a previously published manuscript (Amalia Luce et al. Oxid Med Cell Longev. 2021) where we demonstrated that PD enhanced osteoblast differentiation markers including alkaline phosphatase activity, osteopontin, and Notch2.  Moreover, the Mesenchymal stem cells were grown, without specific inducing differentiation factors, on both PCL and PD-PCL fibers to investigate the ability of the delivering and slow release of PD from PD-PCL scaffold to promote osteogenic cell differentiation. The osteogenic bone differentiation of PD-PCL scaffolds was evaluated, by morphological change and alkaline phosphatase activity.  We have now added this information in the “Discussion” section of the revised version of the manuscript.

Issue 4. Scale bars in Figs. 4-6 are unclear.

Response: As correctly suggested by the referee, we have corrected the scale bars in figures 4-6.

Issue 5. 5 is dislocated.

Response: As correctly suggested by the referee, we have centered the figure 5.

Issue 6. In Fig. 6, cell number on PCL seems less than that on PD-PCL on day 7, which is disagreed with results in Fig. 3. Can authors stain both ALP and COL I in the merge image to see the stage of differentiation?

Response: We thank the reviewer for her\his kind comment.  The results reported in figure 3 is a quantitative analysis about cell survival. On the other hand, confocal microscopy images, in figure 6, are representative of localized field to show the Col1 immunopositivity.

Issue 7. Page 9, how could authors comment that the good uptake and slow release of PD?

Response: PD-PCL nanofibers show a time-dependent release of active polydatin moieties. In fact, we found PD up-to 7 days in the medium of cells seeded on PD-PCL. This effect is due to the porous nature of the PCL nanofibers and to the interaction between PD and PCL with possible creation of hydrogen bonds within PD and PCL that allows the release of PD from the scaffolds due to their weak binding. This information is shown and commented in Figure 1.

Reviewer 2 Report

  1. Please put figure 5 in the visible position.
  2. The resolution of figure 6 needs to be improved.

Author Response

Response to Reviewer 2

Issue 1. Please put figure 5 in the visible position.

Response: As correctly suggested by the referee, we have put the figure 5 in the visible position.

Issue 2. The resolution of figure 6 needs to be improved.

Response: As correctly suggested by the referee, we have increased figure 6 resolution in the revised version of the manuscript.

Round 2

Reviewer 1 Report

It seems more acceptable now.